# Metagenomic Assessment of DNA Viral Diversity in Freshwater Sponges, *Baikalospongia bacillifera*

**DOI:** 10.3390/microorganisms10020480

**Published:** 2022-02-21

**Authors:** Tatyana V. Butina, Ivan S. Petrushin, Igor V. Khanaev, Yurij S. Bukin

**Affiliations:** Limnological Institute, Siberian Branch of the Russian Academy of Sciences, 664033 Irkutsk, Russia; ivan.kiel@gmail.com (I.S.P.); igkhan@lin.irk.ru (I.V.K.); bukinyura@mail.ru (Y.S.B.)

**Keywords:** freshwater sponges, sponge holobionts, DNA viruses, viral diversity, metagenomics, viromes, gene prediction, functional analysis, Lake Baikal

## Abstract

Sponges (type Porifera) are multicellular organisms that give shelter to a variety of microorganisms: fungi, algae, archaea, bacteria, and viruses. The studies concerning the composition of viral communities in sponges have appeared rather recently, and the diversity and role of viruses in sponge holobionts remain largely undisclosed. In this study, we assessed the diversity of DNA viruses in the associated community of the Baikal endemic sponge, *Baikalospongia bacillifera*, using a metagenomic approach, and compared the virome data from samples of sponges and Baikal water (control sample). Significant differences in terms of taxonomy, putative host range of identified scaffolds, and functional annotation of predicted viral proteins were revealed in viromes of sponge *B. bacillifera* and the Baikal water. This is the evidence in favor of specificity of viral communities in sponges. The diversity shift of viral communities in a diseased specimen, in comparison with a visually healthy sponge, probably reflects the changes in the composition of microbial communities in affected sponges. We identified many viral genes encoding the proteins with metabolic functions; therefore, viruses in Baikal sponges regulate the number and diversity of their associated community, and also take a part in the vital activity of the holobiont, and this is especially significant in the case of damage (or disease) of these organisms in unfavorable conditions. When comparing the Baikal viromes with similar datasets of marine sponge (*Ianthella basta*), in addition to significant differences in the taxonomic and functional composition of viral communities, we revealed common scaffolds/virotypes in the cross-assembly of reads, which may indicate the presence of some closely related sponge-specific viruses in marine and freshwater sponges.

## 1. Introduction

Sponges (type Porifera) are unusual representatives of Metazoa, striking with a variety of shapes, colors and structures. These invertebrates are widespread geographically, and some of them are endemic in several locations [1,2]. Sponges accommodate in their bodies various microorganisms such as heterotrophic bacteria, cyanobacteria, microscopic algae, archaea, dinoflagellates, fungi, and viruses [3]. Associated communities of sponges (holobionts) have unique properties—high diversity, abundance and biomass, contribution to primary production and nitrification through complex symbiosis, high chemical and physical adaptation, biomineralization, water filtration, etc.—which determine the important role of sponges and make them an integral part of marine and freshwater ecosystems [4,5,6]. Extracts of various types of sponges have high antibacterial and antiviral activity against many serious pathogens of animals and humans, mainly due to the metabolites of associated symbionts [7]. This also makes the sponges especially valuable and attractive to researchers [8].

Currently, more than 8400 species of sponges are known around the world, and only a small number of them (about 238 sponge species) live in fresh waters; they include several families of the order Spongillida, the class Demospongiae [9]. Two families represent Baikal sponges: endemic Lubomirskiidae (includes 4 genera and 15 species) and cosmopolitan Spongillidae (3 genera and 5 species) [9,10,11]. The endemic species, *Baikalospongia bacillifera*, having a massive globular shape, is one of the most widespread in Lake Baikal [12].

Viruses are the least studied component of the sponge community, which is explained by the methodological difficulties in the investigation of the existing variety of viruses. However, new technologies and approaches such as next-generation sequencing and metagenomic analysis have become good alternatives to classical virological methods and useful aids in the study of uncultivated viruses [13], including the viral communities of sponge holobionts [14,15,16].

The first detection of virus-like particles (VLPs) in marine sponges, which were similar in morphology to adenoviruses, dates back to 1978 [17]. Later, picornaviruses and mimiviruses were detected in some sponges [18,19]. Recently, a large number and variety of viruses in associated communities of sponges have been discovered during a mass electron microscopic (TEM) study [20] and confirmed by metagenomic sequencing of viral communities from various marine sponge species. Such virome studies were carried out for sponges inhabiting the Great Barrier Reef [21,22], the area of the Southwestern Atlantic Ocean (Arraial do Cabo Bay, South-Eastern Brazil) [23], and Lake Baikal [24,25]. Some studies were focused on the study of RNA viruses in sponges [26,27]. In general, various virome sequences related to viruses from more than 20 families, as well as many unclassified and unidentified viruses, were revealed in different marine and freshwater sponges [14,15,16,25]. The bulk of the sponge-associated viruses are DNA viruses—both single-stranded (ssDNA) and double-stranded (dsDNA) ones [21]. A significant host species specificity for viruses infecting sponge holobionts was shown both at the taxonomic and functional levels [14,22,23]. Thus, studies concerning the composition of viral communities in sponges have appeared rather recently [3,28], and many questions regarding the diversity and participation of viruses in the functioning of the sponge holobiont remain largely undisclosed. To the best of our knowledge, the viral communities of freshwater sponges (except for Baikal ones) have not yet been studied.

Viruses most likely play a significant role in the associated community of sponges inhabiting Lake Baikal and affect the ecology and the general state of the lake as a whole. Lake Baikal, like other freshwater bodies, experiences a certain level of anthropogenic pressure; global climate changes also affect its condition. In recent years, anomalous phenomena have occurred on Lake Baikal, which were most obvious in coastal recreational areas of the lake. Changes are observed in the composition of benthic communities, among them: the rich development of filamentous algae and cyanobacteria, overgrowth of macrophytes with ciliates, and massive damage and death of the sponges [12,29,30,31,32]. The problem of sponge disease concerns not only Baikal sponges; cases of their visible changes and damage were described all over the world [33,34,35]. Understanding the cause of sponge diseases is still insufficient. Most often, there are significant changes in the quantitative and qualitative composition of microorganisms in diseased sponges [33,35,36,37,38,39,40]. Sometimes, the pathogenic bacteria are present, and they probably participate in epizootics [41]. For marine sponges, the root cause can be an increase in water temperature [42]. The generally accepted opinion is that sponge diseases (or syndromes) may be due to the disruption of complex interactions within the holobiont under environmental stress, resulting in opportunistic or polymicrobial infections [34,35,42]. The role of viruses in sponge disease is largely unexplored. This topic requires special attention and great efforts of researchers.

The aim of this study was to assess the diversity of DNA viruses in the associated community of the Baikal endemic sponge, *Baikalospongia bacillifera*, (visually healthy and damaged) using a metagenomic approach (the data reported in [25]), and to compare the virome data from the samples of sponges and Baikal water (control sample, [43]).

## 2. Materials and Methods

### 2.1. Sampling and Sample Processing

The *B. bacillifera* sponges were sampled in sterile tubes in the southern basin of Lake Baikal, near Bolshiye Koty (51°54′07.5″N, 105°06′12.0″E), at depths of about 16 m in May 2018 by divers using lightweight diving equipment. The two specimens of *B. bacillifera* of 5–7 cm^3^ in volume were collected and used in this study: one looked healthy (Sv2478.2h), and another had necrosis lesions (Sv2475.1d) (Figure 1). Sponge samples were processed, and concentrates of VLPs were obtained as described in [25].

At the same time, the control near-bottom water samples were also taken from the sponge sampling site at depths of 10, 12, and 15 m. The sampling was carried out by a diver using a bathometer. The water samples were filtered through 0.2 mm nitrocellulose filters (Sartorius, Goettingen, Germany) and combined (sample Lbw.4g). The filtrate containing virus-like particles was concentrated as described in [43].

Detailed description of sample processing (of the sponges and water), VLPs and viral DNA extraction, further sequencing, and bioinformatics analysis can be found in the Appendix A.

### 2.2. Library Preparation and Sequencing

The preparation and sequencing of DNA libraries were performed in The Center of Shared Scientific Equipment “Persistence of Microorganisms” of the Institute for Cellular and Intracellular Symbiosis, Ural Branch of the Russian Academy of Sciences, Orenburg, Russia. Sequencing of the libraries was conducted on the MiSeq platform (Illumina, San Diego, CA, USA) using MiSeq Reagent Kit v3 (2 × 300cycles).

Unprocessed virome reads for samples Sv2475.1d, Sv2478.2h, and Lbw.4g were submitted to the National Center for Biotechnology Information (NCBI), Sequence Read Archive (SRA) database (BioProject PRJNA577390, BioSamples SAMN13025046, SAMN13025227, and SAMN16330433) [25,43]. The direct URL to the data is as follows: https://www.ncbi.nlm.nih.gov/sra/PRJNA577390 (accessed on 20 December 2021).

### 2.3. Initial Shotgun Metagenomic Data on DNA Viruses in Marine Sponges and Water Samples

For comparative analysis, we also used the NCBI SRA datasets on marine sponge *Iantella basta* (Pallas, 1766) (class Demospongiae) and ocean water viromes (Great Barrier Reef (GBR), Davies Reef, sampled in January 2014; [22]) sequenced using the same library preparation and sequencing techniques as in our study (the Illumina MiSeq platform). Similar to our data, in the study of the sponges *I. basta*, diseased and healthy specimens were sampled, and at the same time, a control water sample was taken at the sponge sampling site, as follows from the description of the marine samples (Table 1).

The paired reads of marine viromes were combined into one FASTQ dataset together with the Baikal ones; then joint primary processing of paired reads was carried out as described below. All data were used for a hybrid metagenomic assembly (cross-assembly) in one round of data analysis.

### 2.4. Primary Processing of Virome Reads

The quality visualization of the virome datasets (paired reads) was carried out using the FASTQC program. Trimming of reads by the quality was carried out with the Trimmomatic V 0.39 program [44].

### 2.5. Assembly of Virome Reads, Identification, and Taxonomic Assignment of Viral Scaffolds

The assembly of viral reads and further taxonomic identification of viral scaffolds were carried out as reported before in the study of water samples from different areas of Lake Baikal [45]. Briefly, the SPAdes 3.13.1 (metaSPAdes) [46] was used for the de novo cross-assembly of datasets (marine and freshwater, Table 1). The scaffolds with coverage more than 5 and a length of ≥5000 bp were used for further analysis. The VirSorter tool [47] was used for identification of the viral scaffolds and open reading frames (ORFs) in them. Taxonomic identification for the viral scaffolds was carried out by comparisons of scaffolds sequences with the NCBI RefSeq complete viral genome and viral proteome database [48] with BLASTp and BLASTn algorithms [49]. A virus taxon from NCBI RefSeq with the highest proportion of coverage in alignments of the nucleotide sequences was chosen as the scaffold virotype identifier.

The Burrows–Wheeler Aligner (BWA) software [50] was used to map paired-end reads on scaffolds and calculate the total coverage of viral scaffolds in the assembly and coverage of scaffolds by reads from each sample. The count table of viral scaffold representation in the analyzed samples normalized to the length of scaffolds was constructed.

### 2.6. Statistical Analysis of Taxonomic Diversity

The potential (underestimated) number of virus scaffolds and virotypes (species richness) in communities was evaluated using Chao1 [51] and ACE [52] indices. Shannon and Simpson indices [53] of biodiversity were also calculated (Table 2) for virus scaffolds and virotypes. The taxonomic composition similarity of the samples (similarity in virus scaffold count table per samples) was visualized using hierarchical cluster analysis with bootstrap support calculation of clustering in the “pvclust” [54] package for the R and the nonmetric multidimensional scaling (NMDS) ordination method. Biodiversity analysis and NMDS were carried out in the “vegan” package for the R [55].

Dominant scaffolds and virotypes in Baikal and marine samples were visualized with the heat map using the “gplots” [56] package for the R. Columns (samples) in the heat map were clustered and grouped in similarity order (i.e., Bray–Curtis distance metric and the complete-link clustering method).

The significance of the difference between the samples in counts of virotype reads was assessed using the chi-square test for independence with Bonferroni *p*-value correction.

### 2.7. Functional Assignment of Viral Communities

Functional assignment of predicted viral proteins (ORFs) was carried out in three different ways: (1) matched ORFs with the UniProtKB/Swiss-Prot database [57] by the BLASTp algorithm; (2) matched ORFs with functional motifs of proteins in the Pfam database [58] using an online resource (https://www.ebi.ac.uk/Tools/pfa/pfamscan/, accessed on 20 September 2021) [59]; (3) matched ORFs with functional motifs of proteins in the KOfam database using an online resource (https://www.genome.jp/tools/kofamkoala/, accessed on 20 September 2021) [60]. All three results were transformed to KO (KEGG pathway classification functional groups) anthologies [61] and processed in the “KEGGREST” package [62] for the R. The count of the predicted viral proteins in samples was transformed into counts of the KEGG pathway classification groups that were normalized for the average number of hits on the viral proteins in each sample. The counts of AMGs (auxiliary metabolic genes) viral proteins in different samples were visualized with a heat map generated using the “gplots” package [56] for the R.

### 2.8. Viral Hosts Prediction

Host prediction for the set of viral scaffolds was carried out by the method as described previously [45], basing on taxonomic identification of predicted viral scaffolds and the Virus–Host database [63]. The count of the viral scaffolds was transformed into tables representing DNA viruses (virotypes) that infect a certain host species and analyzed in the NMDS scatter plot of viral scaffolds count table comparisons.

### 2.9. Bacterial Defense Mechanisms against Viruses

The genomic assemblies of two bacterial strains, *Flavobacterium* sp. SLB02 and *Janthinobacterium* sp. SLB01, isolated from the diseased Baikal sponge *Lubomirskia baikalensis* has been published previously [64]. In our study, we analyzed whether these strains have any antiviral defense mechanisms. To search for defense mechanisms, we analyzed bacterial genomes using the Prokaryotic Antiviral Defense LOCator (PADLOC) [65] and CRISPRCasFinder [66] algorithms, and the numbers of CRISPR arrays were detected. Spacers’ sequences were aligned with viral scaffolds from our study using the BLASTn-short algorithm [49], as recommended in [67].

## 3. Results

### 3.1. Taxonomic Affiliation of Viral Sequences in Baikal Samples

After processing and filtering the raw reads, we obtained the three sets of Baikal virome data, from 3.6 to 9.5 million paired reads each (Table 2). The proportions of viral reads in our datasets ranged from 15.3% to 28.8%; those of bacterial and eukaryotic sequences did not exceed 15% and 2.4%, respectively, while a significant part of the reads was not identified (up to 73.3%) (Appendix A). The percentage of viral reads was found to be comparable with that of other virome studies with viral particle enrichment (or even higher, such as compared with the marine samples we used in this study, Table 1).

In total, 2916 scaffolds with length ≥ 5000 bp and coverage ≥5 were assembled using the metaSPAdes software together with marine virome reads; 673 scaffolds were identified as viral by the VirSorter program, more than 400 scaffolds consisted of Baikal reads. The total numbers of Baikal reads belonging to identified viral scaffolds were 637,148 (14.7% of the total reads, sample *B. bacillifera*_Sv2475.1d), 681,061 (19.1%, sample *B. bacillifera*_Sv2478.2h), and 805,244 (8.5%, sample Baikal water_Lbw.4g). For the marine sponge viromes [22] and the ocean water sample (Table 1), the percentages of viral reads were 4.7% to 11.8% (Table 2).

Taxonomic affiliation (as similar viral genome or virotype) was assigned for 318 to 338 (78.7% to 79.0%) scaffolds (Table 2; Appendix A). In total, more than 168 virotypes were identified in the samples of the *B. bacillifera* sponges and surrounding water. The largest number of virotypes (183) was found in the Baikal water sample, and the smallest number (163) was found in the diseased sponge (Sv2475.1d). In general, the indices of diversity and species richness of the Baikal viral communities were high and comparable with or higher than those for marine samples (Table 2). The greatest variety of viruses was observed in control water samples with both fresh and sea water.

The identified virotypes belonged to 11 families of DNA-containing viruses (Figure 2a; Appendix A). The most numerous of them were the tailed bacteriophages of the families *Siphoviridae* (35.8% to 45.2% of viral reads), *Podoviridae* (20.2% to 30.6%), and *Myoviridae* (6.6% to 12.1%). Additionally, bacteriophages of the families *Herelleviridae* and *Ackermannviridae* were also detected but in small numbers (less than 0.3%). Virophages of the family *Lavidaviridae* were among the dominant ones (3.1% to 6.3%). These satellite viruses infect protozoa but only in the case of co-infection with other viruses, members of the family *Mimiviridae* [68]. The mimiviruses also presented a small fraction of viral scaffolds. Among the others, there were the viral families, known representatives of which infect microalgae (*Phycodnaviridae*), archaea (*Bicaudaviridae*), and arthropods, including crustaceans (*Baculoviridae*, *Poxviridae*) and vertebrates (*Poxviridae*). Notably, a large number of sequences identified as viral using the VirSorter program had no analogues in the NCBI database (14.5–19.0% of unknown reads) (Figure 2a; Appendix A).

The dominant and other families were similar for all Baikal samples, but their proportions differed. For example, the *Siphoviridae* prevailed in the Baikal water sample, but the *Podoviridae*, as well as virophages of the *Lavidaviridae* family, were more represented in the sponge samples (Figure 2a; Appendix A). The differences in the representation of viral families and virotypes in the Baikal samples are summarized in Table 3.

### 3.2. Virotypes Diversity

Differences between the freshwater samples are clearly observed during the comparison of the sets of the identified virotypes (Figure 2b; Appendix A). In the Baikal water sample, the *Arthrobacter phage Decurro*, the *Synechococcus* phages *ACG-2014h*, *S-SKS1,* and other cyanophages, *Xylella phages Sano* and *Xfas53,* were the most abundant. The sponge samples differed from the control water sample by a high content of sequences that corresponded to genomes of the *Cellulophaga phages phi10:1*, *phi38:1,* and *phi19:3*, *Yellowstone lake virophage* 6, and others (Figure 2b; Appendix A). In the sample of a diseased sponge, compared with others, there was higher content of sequences similar to the *Enterobacteria phage Sf101*, *Bdellovibrio phage phi1422,* and *Croceibacter phage P2559Y*. In the sample of a healthy sponge, the *Nonlabens phage P12024S*, *Gordonia* phages (*GMA1*, *Wizard*, etc.), and *Synechococcus phage S-CBS2* prevailed.

Among eukaryotic viruses, *Melanoplus sanguinipes entomopoxvirus*, the *Yellowstone lake phycodnavirus* 1 and 2, *Paramecium bursaria Chlorella virus* 1, and *Bathycoccus* sp. *RCC1105 virus BpV1* from the family *Phycodnaviridae* were detected in the Baikal sponges and near-bottom water; the scaffolds similar to the virotypes *Paramecium bursaria Chlorella virus* 1 and *Yellowstone lake phycodnavirus* 1 were mainly covered with reads from sponge samples, but *Yellowstone lake phycodnavirus* 2 was covered with reads from a control water sample (Appendix A).

### 3.3. Putative Range of Viral Hosts

We defined the potential host range for the detected Baikal viruses (virotypes) based on known hosts for the identified virotypes according to the Virus–Host database [63]. Using this approach, the putative hosts were determined for a wide range of viruses (bacteriophages and others). Unfortunately, other available resources and software predict mainly prokaryotic hosts. In total, six bacterial, two archaeal, and four eukaryotic phyla, as well as one viral family (the *Mimiviridae* affected by virophages in case of coinfection of protists) were identified (Figure 2c; Appendix A). Among the bacterial taxa, the Bacteroidetes, Actinobacteria, Proteobacteria, and Cyanobacteria prevailed (up to 21.5–38.8% in the samples, with the mean value of more than 13.4% in terms of the number of reads), while Firmicutes and Verrucomicrobia did not exceed 4.3% and 0.02% of the reads, respectively (the mean value less 2.4% and 0.01% of the reads) (Appendix A). A significant difference in the range of viral hosts was observed between the samples. For example, the Bacteroidetes were much more abundant (about an order of magnitude) in sponge samples than in the water sample (32.4% to 38.8% vs. 3.1%), while the Cyanobacteria and Firmicutes predominated in Baikal water (26.3% vs. 6.0% to 8.0% and 4.3% vs. 1.3% to 1.7%, respectively) (Appendix A). Predicted eukaryotic hosts were generally few (less than 0.8%); among them, the Amoebozoa and Chlorophyta were the most represented.

Analysis of the bacteria *Janthinobacterium* sp. strain SLB01 and *Flavobacterium* sp. Strain SLB02 from the Baikal sponge *Lubomirskia baikalensis* [64] revealed seven variants of CRISPR-Cas spacers (one in SLB01 and six in SLB02) and five other types of antiphage defense systems, including recently discovered [65,69] (dGTPase, Zorya_type I, Septu_type I, Gabija, Cbass_type III) (Appendix A). The 14 matches of CRISPR-Cas spacers from Strain SLB02 and SLB01 were found with 13 different viral scaffolds from the sponge B. bacillifera (Appendix A). Taxonomically, these scaffolds belong to 12 different virotypes (viruses of bacteria, cyanobacteria, or eukaryotic algae). The length of complete matches between spacer and the viral genome ranges from 7 to 15 nucleotides, which is sufficient to counteract the virus. This analysis suggests that some similar bacteria and viruses associate with the sponges *L. baikalensis* and *B. bacillifera* (healthy or diseased).

### 3.4. Functional Analysis of Baikal Viromes

We carried out a functional analysis of viromes and identified 5 main and 29 secondary categories of proteins according to the KEGG Orthology database (Figure 3; Appendix A). A total of 15,453 open reading frames were predicted in viral scaffolds using the VirSorter program, among which 2629 proteins (proteins with KO_ID) were identified using the Pfam, KOfam and UniProt databases (2456, 514, and 402, respectively). Most of them (1245) were proteins with an undefined functional category (unclassified or “no type” in Appendix A); they were not considered in further analysis.

The largest number of identified proteins in *B. bacillifera* sponge viromes belonged to the “Metabolism” and “Genetic Information Processing” main categories (Figure 3). In the first category, the largest number of reads in both sponges (healthy and diseased) belonged to proteins involved in the metabolism of nucleotides, cofactors and vitamins. Additionally, the proteins of amino acid metabolism predominated in the diseased sponge. The proteins of the ‘Replication and repair’ prevailed in the Baikal water sample (LBw.4g), but the ‘Folding, sorting and degradation’ and ‘Translation’ (of the main category “Genetic Information Processing”), as well as the ‘Cell growth and death’ (“Cellular Processes”), were the most abundant in sponges. In general, the functional profile in the two sponge samples was similar, apart from the proteins of the secondary category ‘Amino acid metabolism’ (much more in diseased sponges) and the ‘Cell growth and death’ (also more represented in the diseased sponge). The proportions of reads in some other functional groups also differed but insignificantly.

In the “Metabolism” category, among the most numerous proteins in terms of diversity and proportion of reads, in addition to proteins involved in the metabolism of nucleotides and amino acids, there were enzymes participating in the metabolism of cofactors and vitamins (Figure 3 and Figure 4). The proteins of folate biosynthesis (thymidylate synthase, 2-amino-4-hydroxy-6-hydroxymethyldihydropteridine diphosphokinase, and others) were the most numerous (Appendix A). Conversely, the proteins of ‘Riboflavin metabolism’ (riboflavin kinase, archaea type) predominated in the diseased sponge. In this secondary category (‘Metabolism of cofactors and vitamins’), we also identified the proteins involved in the metabolism of biotin, nicotinate and nicotinamide, thiamine, porphyrin, and chlorophyll. In terms of the number of reads, the most numerous were also the proteins of the biosynthesis and metabolism of glycans. Among the enzymes of biosynthesis of secondary metabolites, we identified those involved in streptomycin, acarbose and validamycin, and staurosporine biosynthesis (Appendix A). The presence and expression of auxiliary metabolic genes (AMGs) for synthesis of vitamins, antimicrobials, and toxin protection in marine sponges have been reported previously [14,15]. In general, in the diseased sponge, the proportion of enzymes of all metabolic categories, except for amino acid metabolism, was slightly lower than in the healthy one (Figure 3; Appendix A).

### 3.5. Comparative Analysis of Freshwater and Marine Viromes

Clustering using UPGMA (unweighted pair group method with arithmetic mean, Figure 5a) and NMDS (non-metric multidimensional scaling; Figure 5b–d) based on the similarities and differences of assembled virome reads (scaffolds) identified three groups of samples. All samples from Lake Baikal were included in the first group, the second consisted of the samples of marine sponges, and the third group was a virome of ocean water from the sampling site of the *I. basta* sponges. Thus, the distances between marine samples (sponges and surrounding water) turned out to be much greater than between the Baikal samples (Figure 5a). The distribution of biplots also showed significant distances between samples of freshwater and marine sponges. The differences in viromes were clearly traced in mainly different directions of vectors of large taxonomic groups of viruses (families of virotypes) (Figure 5b), their predicted hosts (Figure 5c), and metabolic functions of viral communities in the samples (Figure 5d).

We compared the taxonomy of virotypes identified in metagenomic data from Baikal and marine (GBR) samples. The *Myoviridae*, *Poxviridae*, *Ackermannviridae*, and *Mimiviridae* generally predominated in GBR samples, especially in the *I. basta* sponges (Appendix A). Also noteworthy is the higher content of the *Microviridae* viruses (especially in plankton—6.4%) and the *Phycodnaviridae* viruses (2.7% to 7.2%) in marine samples (Appendix A). The *Siphoviridae*, *Podoviridae*, *Lavidaviridae,* and *Baculoviridae,* on the contrary, reliably prevailed in the Baikal samples. In general, the composition and proportions of the dominant families in the samples of water and sponges differed, as well as differing between the healthy and diseased sponges in the marine ecosystem, in contrast with the Baikal one.

The list of dominant virotypes for marine and Baikal samples mostly did not overlap (Appendix A). For example, the cyanophages dominating in marine viromes (*Prochlorococcus phage P-TIM68*, *Synechococcus phage S-WAM2*, *Synechococcus phage S-CAM22,* and others) were minor in the Baikal samples, and they were mainly represented by the *Myoviridae* family, in contrast with the Baikal ones, where cyanophages of the families *Siphoviridae* and *Podoviridae* were also present. The algae viruses *Ostreococcus lucimarinus virus* 7 and *Ostreococcus tauri virus* 1 were also among the most numerous virotypes in marine sponges, and the *Emiliania huxleyi virus* 86 predominated in seawater. All of them also had low representation in the Baikal samples. Despite the obvious differences between marine and freshwater sponge samples, we also identified a number of common scaffolds/virotypes, such as the dominant *Cellulophaga* phages (*phi38:1*, *phi10:1*, and *phi19:3*), some *Synechococcus* phages (*S-SKS1* and others), *Yellowstone lake phycodnaviruses* 1 and 2, and others, as well as a number of unidentified sequences (Appendix A).

In marine samples, as well as in Baikal ones, bacteria of the phyla Bacteroidetes and Arthropoda (insects) prevailed as the hosts in the sponge samples, while the Proteobacteria Actinobacteria, Firmicutes, and archaea Euryarchaeota predominated in the water sample (Appendix A). In contrast with the Baikal samples, cyanobacteria prevailed in the *I. basta* sponges (but not in seawater). The number of Proteobacteria in diseased and healthy sponges remained almost the same. Furthermore, in the list of hosts of marine viruses, we identified the bacteria Chlamydiae (microviruses), widespread ocean unicellular Haptophyceae algae (*Emiliania huxleyi*), and *Escherichia virus P2* (the *Myoviridae* family) that helps for the lytic growth of satellite *Enterobacteria P4* (infects *E. coli* and other Enterobacteriaceae).

The functional profiles of all the samples of the *I. basta* sponge were similar. A significant difference was observed only in the ‘Cell growth and death’ category—namely, an increase in proteins of this group in the healthy sponge specimen (Figure 3). The profile of the control seawater sample was very different from the distribution of proteins in the *I. basta* sponges, as in the case of the Baikal samples. Incidentally, the enzymes of replication and repair predominated in sea water, as well as in fresh water; however, in general, we did not observe correlations in the distribution of functional categories and in the differences between the samples of sponges and surrounding waters in freshwater and marine ecosystems (Figure 3; Appendix A).

## 4. Discussion

### 4.1. Analysis of the Reads Assembly of Marine and Freshwater Virome Samples

In this study, we examined the diversity of the DNA viral communities in two individuals of the Baikal endemic sponges, *Baikalospongia bacillifera* (tentatively healthy and damaged with necrotic lesions) [25], by assembling metagenomic reads and describing the viral scaffolds and predicted proteins. For comparative analysis, we selected from the NCBI SRA database a similar set of virome data from non-diseased and diseased (with unspecified syndrome) marine sponges, *Ianthella basta*, sampled from the Great Barrier Reef [22], as well as of the water sample from the sponge sampling site (Table 1).

To analyze the virome data on Baikal and non-Baikal samples (Table 1), as well as in our previous study of viruses from aquatic communities [45], we applied the approach of the de novo cross-assembly of metagenomic reads. All original virome reads were combined into one sample (all left reads into one sample of left reads, all right reads into one sample of right reads). Thereafter, the resulting set of right and left reads was used for the cross-assembly and identification of viral scaffolds. This approach has an advantage over assembling each sample separately because it allows the comparison of viromes of different samples without direct linking of the resulting scaffolds to the taxonomy of viruses. By mapping the initial reads of each metagenomic sample to the resulting scaffolds, we estimated the representation of each scaffold in the sample, thereby, obtaining a table of quantitative data for direct analysis using computational methods of environmental research. The cross-assembly enabled us to correctly use the scaffolds that were not identified as virotype for the comparative analysis of samples. Another advantage of cross-assembly is the ability to assemble longer virus scaffolds with an increased dataset owing to the presence of genetic material of the same or closely related virus species in different samples. A disadvantage of this cross-assembly approach is the need to use a supercomputer with a very large amount of RAM (1 TB or more) to process a large k-mer array required to construct a De Bruijn graph in the genomic data assembly algorithm [70].

Of the total data, about 23% of the scaffolds were identified as viral, and most of these scaffolds (about 76%) were similar to the fragments of genomes or proteomes from the NCBI RefSeq viral database. The largest number of assembled viral sequences had low similarity with known viral genomes (considering genome-scaffold coverage, number of similar proteins, and sequence similarity). Most of the viral sequences had genome coverage within 10% to 20%, rarely more than 40%. From 5% to 19% of the original metagenomic reads mapped to scaffolds were identified as viral (Table 2). Thus, most of the genomic reads were unidentified, and most predicted protein-coding genes had no similarities with protein motifs from the databases used for the analysis. Such results are typical for virome studies; in other analyses of viral communities in aquatic [45,71,72,73,74] and other environments [75,76], including those associated with marine sponges [15], the percentage of viral reads was even lower (a few percent).

### 4.2. The Diversity of Viral Communities in the B. bacillifera

In Baikal virome datasets, the identified viral scaffolds were similar to 11 families of DNA-containing viruses. The highest numbers of virotypes belonged to the families of bacteriophages (*Siphoviridae*, *Myoviridae*, *Podoviridae*, and others). This is expectable because the abundance of bacteria accounts for up to 35% of the total sponge biomass, and their densities exceed 10^9^ cells per cubic centimeter of sponge [77]. A study [20] assumed that tailed VLPs may be more abundant on the external surface of the sponges. We also identified the viral families that are known to infect microalgae (*Phycodnaviridae*), archaea (*Bicaudaviridae*), protozoa (*Mimiviridae* and *Lavidaviridae*), and invertebrates (*Baculoviridae* and *Poxviridae*). These families, except for the *Lavidaviridae*, were found earlier in the viromes of marine sponges [14,16]. A large abundance of virophage sequences was discovered previously in the viromes of Baikal water [45].

The greatest similarity of viral scaffolds (based on the similarity percentage and the number of matching proteins) was found with cyanophages, as well as with some viruses that infect Proteobacteria: *Idiomarinaceae phage 1N2-2*, *Pseudomonas phage PA11,* and *Bordetella virus BPP1*. Perhaps, the related viruses, like their hosts, have a wide range of habitats (sea and fresh waters, soils, wetlands, and others). For instance, bacteria of the family Idiomarinaceae (Gammaproteobacteria) were isolated from saline habitats [78]. The bacteria *Pseudomonas* sp. are present in all habitats, but some species such as *P. aeruginosa* tend to be present in areas closely associated with human activities [79]. Representatives of the genus *Pseudomonas* and their viruses were isolated previously from Lake Baikal [80,81]. For eukaryotic viruses, there was the greatest overlap of scaffolds with *Yellowstone Lake virophage 5* and *Melanoplus sanguinipes entomopoxvirus* also isolated from the geographically distant and distinctive environments [82,83].

### 4.3. Comparative Analysis of Viromes of Diseased and Healthy B. bacillifera and the Surrounding Baikal Water

A comparative analysis of the diseased and the visually healthy sponges revealed a greater number of virotypes in a healthy sponge (183 vs. 163). This difference in diversity is most likely associated with a decrease in the total number of associated microorganisms in the necrotic sponge. We determined the same composition but different percentages of viral families in diseased and healthy sponges (Figure 2a; Table 3; Appendix A). For example, the abundance of *Myoviridae* phages (larger number in a diseased sponge) and viruses of algae *Phycodnaviridae* (on the contrary, larger number in a healthy one) varied significantly. The dominant virotypes in the two sponges also differed (Figure 2b; Table 3; Appendix A). The diversity shift in the sponge-associated microbial communities in the unhealthy individuals was reported previously, but different changes were found in analyzed diseased specimens, and no patterns were revealed [35,36,37,38,39,40,84,85]. During the experimental thermal stress, the viral compositions of the *Rhopaloiedes odorabile* sponges from the Great Barrier Reef also changed; this, for example, led to the loss of ssDNA viruses [86].

The comparison of the viromes of the sponges and the control water sample revealed even more significant differences in the composition of viral families, especially, of virotypes (Appendix A; Figure 2a,b). The most significant difference was the much higher abundance of scaffolds similar to picocyanobacterial viruses in the water sample and the high content of viral sequences related to *Cellulophaga* phages in the *B. bacillifera* sponges. Our previous study of the Baikal endemic sponges *Lubomirskia baikalensis* based on marker phage genes g20 also supported the specific viral communities within sponge holobionts [87]. The specificity of sponge viral communities was also shown in the studies of marine sponges [14,22].

We assessed the functional potential and metabolic genes of the viral communities of Baikal sponges and revealed differences in the functional profiles of healthy and diseased individuals as well as in sponges and surrounding water (Figure 3 and Figure 4). In the diseased *B. bacillifera* sponges, functional genes associated with amino acid metabolism increased. Perhaps, this is due to the active recovery processes taking place in the damaged (with lesions of necrosis in this case) sponges. According to our observations, Baikal sponges can heal the affected areas: on some apparently healthy *B. bacillifera* sponges with an intact surface, a violation of the globular structure and large notches were observed during diving and sampling. Moreover, in the viral communities of diseased sponges, the number of genes for biosynthesis of folate and glycans and other metabolic functions (except for ‘amino acid metabolism’) was reduced in the diseased sponge compared with the healthy one (Figure 3; Appendix A). This may be due to a decrease in the taxonomic diversity of viruses in diseased sponges. As a consequence, the important functions that are characteristic of the normal functioning of the sponge holobiont are lost during dysbiosis. Under the conditions of a short-term experiment, the authors [86] did not observe the changes in the functional set of viral genes. However, our data indicate that under natural conditions, with prolonged stress or disease, the shift in the taxonomic composition and metabolic profiles occurs not only in microbial [37,38,39,40,88,89,90] but also in viral sponge-associated communities. Most likely, the recorded changes depend on various factors, and only a more targeted analysis of a large number of sponge samples can yield more accurate conclusions.

Our results reveal the significant differences in the functional potential of viral communities and in the sets of viral metabolic (AMG) genes in the sponges and plankton of Lake Baikal. Thus, the functional analysis also confirmed the nonrandom diversity (i.e., specificity) and the functional role of the viral communities in sponge holobionts.

### 4.4. Putative Viral Hosts for Baikal Viruses

The taxonomic groups of potential bacterial hosts of identified virotypes in the Baikal samples were the Bacteroidetes, Actinobacteria, Proteobacteria, Cyanobacteria, Firmicutes, and Verrucomicrobia (Figure 2c and Figure 5c; Appendix A). These bacterial phyla are known components of sponge microbiomes [3,77]. They were also identified in different Baikal sponges (*L. baikalensis*, *Baikalospongia intermedia,* and *Swartschewskia papyracea*) [37,38,39,40,88,89] and in water column of Lake Baikal [91,92] during previous studies. Representatives of the Bacteroidetes and Actinobacteria (the main groups of predicted viral hosts) are well-known biomass destructors in a wide spectrum of niches, including freshwater [93]. Actinobacteria dominate in microbial community of Lake Baikal, particularly in the coastal zone (from 32% to 69% sequences in the 16S rRNA gene analysis) [90]. In healthy sponges, the Actinobacteria varied over a wide range (from 3.5% to 14%) [88,89]; however, in damaged specimens of *L. baikalensis*, their proportion reached only 8.6%, and their diversity shifted towards the benthic- and soil-derived representatives of the phylum [90].

Recently, the various bacterial strains (35 bacterial ones) belonging to the phyla Actinobacteria, Firmicutes, Bacteroidetes, and Proteobacteria were isolated from symbiotic community of sponge *L. baikalensis* [64,94]; a potential ability of many cultured microorganisms from different taxonomic groups to produce secondary metabolites was also shown [94]. As is known, these specialized compounds are not required for normal cell growth, but they may play an important ecological role in the interactions with surrounding organisms or the environment. We found the multiple viral genes for the synthesis of antimicrobial and other auxiliary metabolites in Baikal sponges. Thus, due to the AMGs, viruses can stimulate the vital activity and increase the competitiveness or communication of bacterial hosts, thereby maintaining the functioning of the entire holobiont, especially under unfavorable conditions and during disease (dysbiosis).

On the other hand, the bacteria are forced to defend themselves against viral infections; there are various defense mechanisms: CRISPR-Cas, restriction-modification (RM), and other systems. In the study of microbial communities of sponge holobionts from deep-sea hydrothermal vents [95], the multiple genes related to diverse anti-phage defense systems (RM, CRISPR-Cas, toxin–antitoxin, and others) were found. We tested the recently published genomes of the bacteria *Janthinobacterium* sp. SLB01 (refers to Proteobacteria) and *Flavobacterium* sp. SLB02 (Bacteroidetes) isolated from diseased *L. baikalensis* sponges [95] and also revealed the presence of various defense systems in them, including CRISPR-Cas; moreover, the matches were found when comparing revealed CRISPR-Cas spacers and virus scaffolds from the sponges *B. bacillifera* (Appendix A).

A new approach (phage fluorescence in situ hybridization-correlative light and electron microscopy, PhageFISH-CLEM) discovered that the phagocytosis of viral particles by sponge cells modulates phage–bacteria ratios and ultimately controls viral infections in Mediterranean sponges. Such tripartite interplay (animal–phage–bacterium) led to dominance of lysogeny the sponge microbiome, while lysis predominates in seawater [96]. However, in our study, the integrases, as temperate phage markers, were revealed mainly in scaffolds from Baikal and GBR waters. In deep-sea hydrothermal vent sponges (in the southern Okinawa Trough), the prophages were also rarely found in the genomes of bacterial symbionts [95], which may indicate distinct phage replication strategies in different sponges or environments.

The Baikal sponges are known to be a host for symbiotic green microalgae [97]. The presence of the Chlorophyta in the range of putative hosts in the samples of *B. bacillifera* (especially in the ‘healthy’ one) (Figure 5c; Appendix A) may be associated with these symbionts previously revealed in the Baikal sponges. As shown, the Chlorophyta gradually disappears in ‘diseased’ samples [39,40] similar to our data. We also notice the presence of the Amoebozoa, which has been poorly studied for the most part in sponge associations. Sponges are inhabited by many other invertebrate species (crustaceans, mollusks, etc.) [98]; this can explain the presence of viruses (virotypes) of the Arthropoda in our data.

### 4.5. Comparative Analysis of Marine and Freshwater Viromes

We also compared the Baikal viromes with similar NCBI SRA datasets (diseased and healthy sponges; sponge and control water samples) from marine sponges *Ianthella basta*, analyzed using the same sample preparation and sequencing methods as in our study. This allowed us to reduce the possible shifts associated with the methodological procedure and increase the chance of a more objective comparison of viral communities of different sponge species.

The number of identified viral scaffolds and virotypes in the viromes of the *I. basta* sponges was lower than in *B. bacillifera* (190 to 225 vs. 404 to 417 scaffolds and 89 to 109 vs. 168 to 171 virotypes), despite a much higher number of reads for the *I. basta* sponges (Table 2). In GBR and Baikal waters, it also slightly differed (384 and 428 scaffolds, 178 and 183 virotypes, respectively). It is important to note that the microbial composition of other freshwater sponges (*Eunapius carteri*, *Corvospongilla lapidosa,* and *Tubella variabilis*) was also more diverse in comparison with the marine sponge microbiota [99,100].

As in the case of Baikal sponges, the number of scaffolds/virotypes in a healthy sponge was greater than in a diseased one. Our analysis revealed that the taxonomic composition of viral communities (at the level of viral families or virotypes), putative hosts, and predicted functional viral genes of marine samples was very different from freshwater samples (Figure 3, Figure 4 and Figure 5). The similarities were observed in the general composition of viral families, some virotypes, and dominant host taxa. The identified common virotypes may indicate the presence of closely related sponge-specific viruses as well as bacterial hosts in marine and freshwater sponges. Interestingly, the shifts in the taxonomic and functional profiles of viral communities in diseased sponges compared with healthy ones are more obvious in Baikal sponges than in marine ones (Figure 5a). Perhaps, this is due to the different degree or duration of damage between GBR and Baikal sponges.

Despite the high content of unicellular algae in Baikal sponges, which gives them a green color [97], and in the lake plankton, viruses of the *Phycodnaviridae*, like other large viruses of the families *Poxviridae* and *Mimiviridae*, were scarce in the Baikal viromes, and their content in the Baikal samples was much lower than in the marine ones. First, the number of bacteriophages presumably really exceeds the abundance of other viruses in Baikal sponges and lake plankton. Second, this pool of viruses may be lost during sample preparation (for example, during centrifugation or filtration, Appendix A). On the other hand, freshwater phycodnaviruses are poorly studied (Baikal phycodnaviruses have not been studied at all), and our list of virotypes contains only the marine *Phycodnaviridae* viruses.

The other significant difference between Baikal and GBR sponges was a much higher abundance of picocyanophage virotypes in the *I. basta* sponges (Appendix A). Cyanophages are most studied in aquatic ecosystems in comparison with other bacteriophages. In spite of this, most of the scaffolds affiliated with cyanophages had low similarity rates, indicating a great diversity and insufficient knowledge about them as well as other bacteriophages in nature. In our study, the cyanophages most similar to the Baikal viral scaffolds (mostly covered by the Baikal virome reads) had a much lower similarity with the sequences from marine samples and vice versa (Appendix A). Such differences were found for many other virotypes, which also indicates a difference in viral communities concerning the composition of analyzed marine and freshwater sponges.

Revealed differences between the freshwater and marine samples can be explained by many factors; the largest and most obvious of them are the difference in sponge species, contrasting habitat (marine and freshwater), and climatic conditions, as well as geographical distance. All of these factors entail a difference in the species composition of the associated sponge community, including viruses; however, it was shown during a global comparison of the microbiota of different sponges that the main structure-forming factor may be the sponge environment (marine or freshwater) [100]. Notably, the unique and extreme conditions of the Baikal ecosystem determine the presence of a large number of endemic species of flora and fauna, including the *B. bacillifera* sponges [101], which also severely affects the formation of microbial communities in Baikal water and sponges.

## 5. Conclusions

This study revealed the high genetic, taxonomic, and functional diversity of the DNA viruses in Baikal endemic sponges *B. bacillifera*. We also identified differences in the composition of viral communities in healthy and diseased sponges, as well as obtaining a large set of viral sequences that did not have similarities with the genomes of known viruses from the NCBI database. Considering that we and other researchers carried out special sample preparation to isolate VLPs and viral genetic material, we can assume that some of the viral sequences belong to unknown viruses that do not have close relatives in the NCBI RefSeq viral database. Since sponge viruses are practically unknown [102], a certain part of the identified or unidentified viral sequences very likely belongs to viruses that infect not only microorganisms associated with sponges but also sponges themselves. We hope that in the future, with the intensive replenishment of international databases with genomic information on the studied viruses, the proportion of viral genetic material identified during the metagenomic analysis will increase. Perhaps the expansion of the databases of protein motifs will also allow for a more informative study of the functional ability of viral communities. A large number of unidentified proteins may belong to AMGs that play an important role in the interaction between the viruses and the hosts at the level of the populations and communities.

## Figures and Tables

**Figure 1 microorganisms-10-00480-f001:**
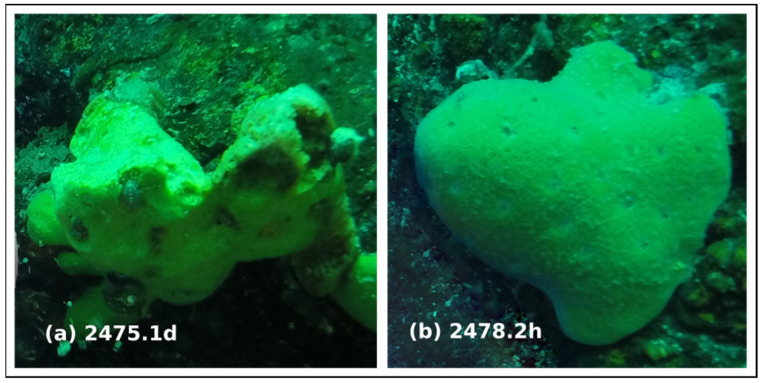
The sponges *Baikalospongia bacillifera*, (**a**) diseased and (**b**) healthy individuals, used in our study of DNA viral communities.

**Figure 2 microorganisms-10-00480-f002:**
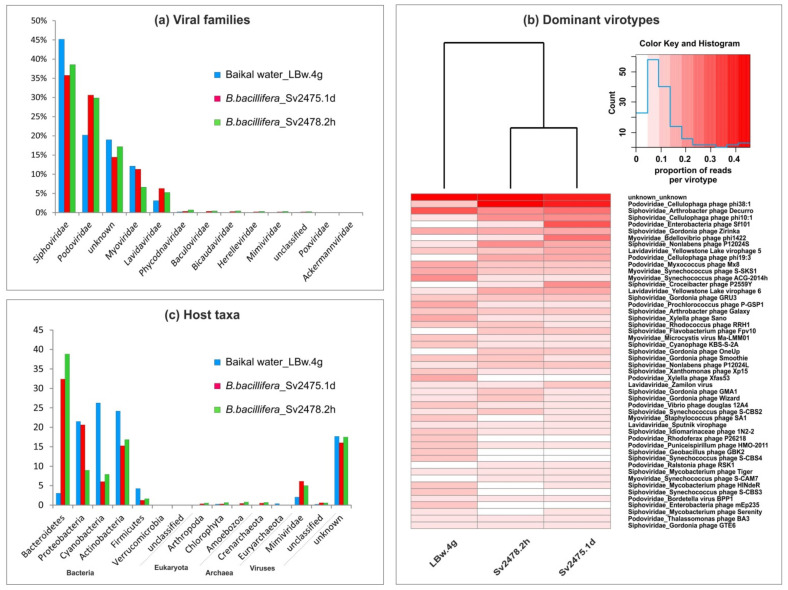
The taxonomic identification of viral sequences and putative hosts for revealed virotypes in the Baikal viromes: (**a**) the percentages of viral families; (**b**) heat maps demonstrating the distribution of 50 dominant virotypes in the samples (clustering was based on the Bray–Curtis distances); (**c**) the predicted viral hosts.

**Figure 3 microorganisms-10-00480-f003:**
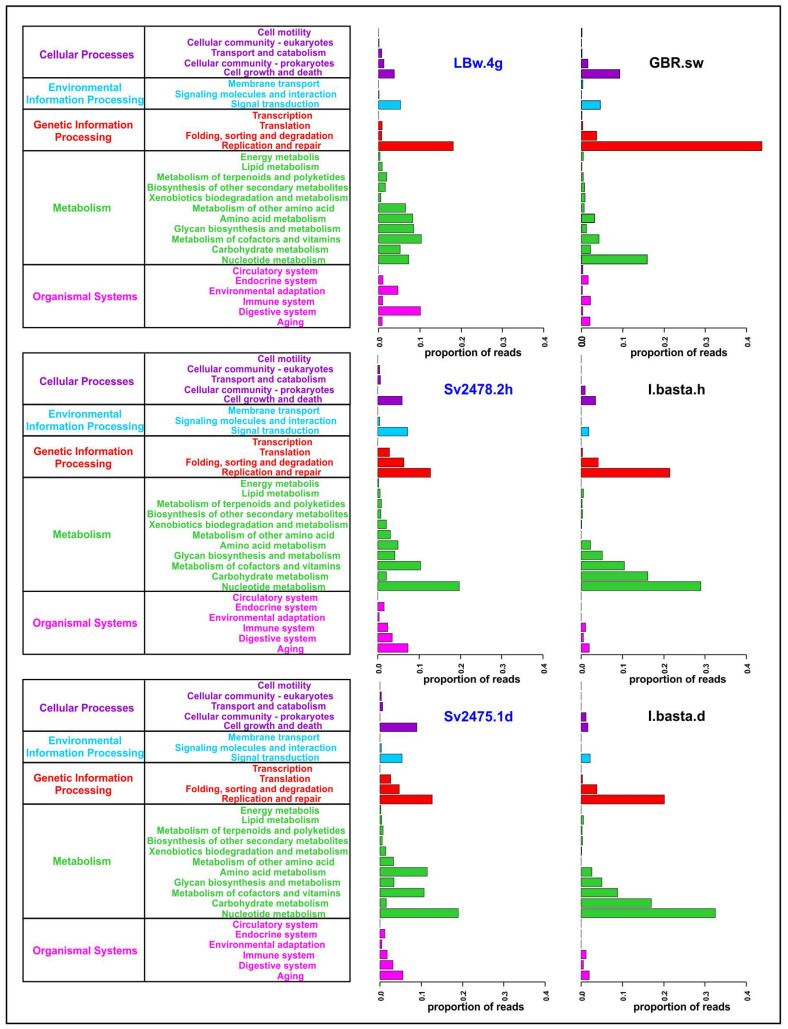
General functional annotation of the sponge and water viromes analyzed in the study (the main and secondary functional categories are indicated, according to the KEGG Orthology). The Baikal samples are highlighted in blue.

**Figure 4 microorganisms-10-00480-f004:**
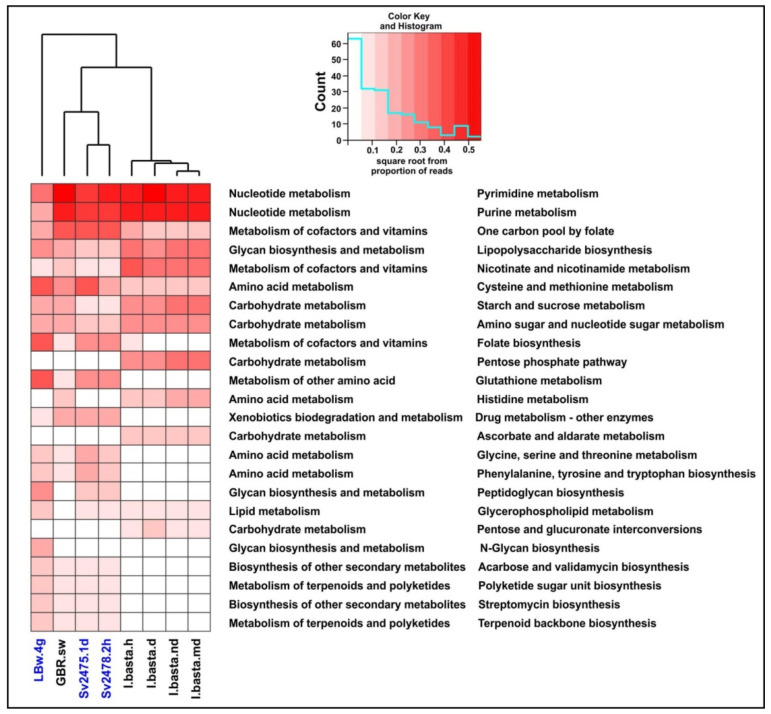
Dominant metabolic functions defined in marine and freshwater virome datasets. The Baikal samples are highlighted in blue.

**Figure 5 microorganisms-10-00480-f005:**
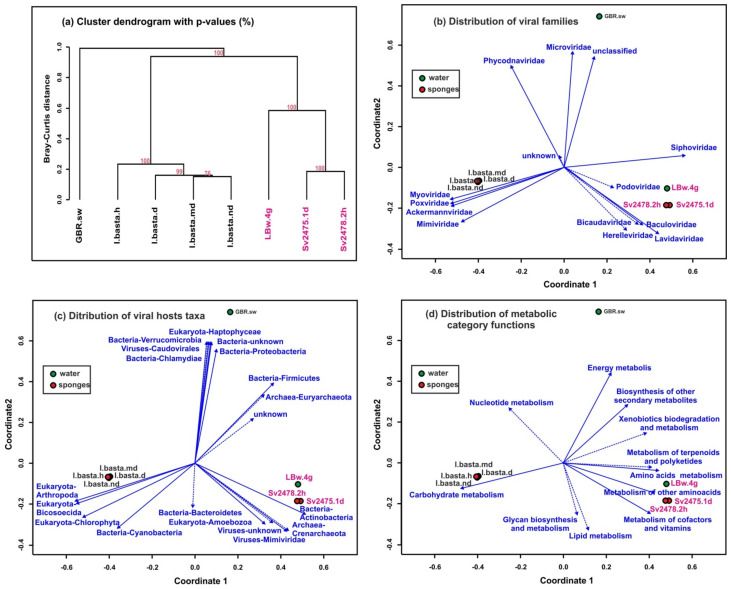
Clustering of samples by similarity of the representation of scaffolds identified as viral. (**a**) Dendrogram constructed by the “average” method based on the Bray–Curtis distances (the dendrogram nodes contain the bootstrap support values). (**b**–**d**) Non-metric multidimensional scaling (NMDS) biplots of the virome datasets showing the following: (**b**) identification of viral taxa by homology with viral genomes and proteomes from NCBI RefSeq (vectors indicate the viral families); (**c**) viral hosts prediction, carried out using the Virus–Host database; (**d**) analysis of metabolic functional categories of viral proteins (AMGs). Unreliable vectors are marked with a dotted line. The Baikal samples are highlighted in pink.

**Table 1 microorganisms-10-00480-t001:** Description of virome datasets used for analysis.

Dataset Name	Sample Description	Geographic Location	Latitude and Longitude	Data	BioProject	Experiments	Reference ^1^
GBR.sw	Seawater control	Australia: GBR, Davies Reef	18.83 S 147.63 E	2014-10	PRJNA388297	SRX2883300, SRX2883301, SRX2883298	- - -
I.basta.h	*Ianthella basta*, disease-free sponge	Australia: GBR, Davies Reef	18.83 S 147.63 E	2014-10	PRJNA388007	SRX2864027, SRX2864026, SRX2864019	- [22] [22]
I.basta.nd	*Ianthella basta*, non-diseased region of diseased sponge	Australia: GBR, Davies Reef	18.83 S 147.63 E	2014-10	PRJNA388007	SRX2864023, SRX2864022, SRX2864016	[22] [22] [22]
I.basta.d	*Ianthella basta*, disease lesion of diseased sponge	Australia: GBR, Davies Reef	18.83 S 147.63 E	2014-10	PRJNA388007	SRX2864021, SRX2864020, SRX2864018	- - -
I.basta.md	*Ianthella basta*, lesion interface of diseased sponge	Australia: GBR, Davies Reef	18.83 S 147.63 E	2014-10	PRJNA388007	SRX2864025, SRX2864024, SRX2864017	- - -
Sv2475.1d	*Baikalospongia bacillifera*, diseased sponge	Russia: Lake Baikal	51.90 N 105.10 E	2018-06	PRJNA577390	SRX6994059	This study
Sv2478.2h	*Baikalospongia bacillifera*, disease-free sponge	Russia: Lake Baikal	51.90 N 105.10 E	2018-06	PRJNA577390	SRX6994055	This study
LBw.4g	Lake Baikal water control	Russia: Lake Baikal	51.90 N 105.10 E	2018-06	PRJNA577390	SRX9228319	This study

^1^ Unpublished datasets are marked with “-”.

**Table 2 microorganisms-10-00480-t002:** General statistics and viral diversity indices for the datasets used in the study.

Samples	Reads_total	Viral Reads in Viral Scaffolds	Viral Scaffolds	Viral Scaffolds with Taxonomic Assignment	Virotypes	Chao1 (Scaffolds/Virotypes)	ACE (Scaffolds/Virotypes)	Shannon (Scaffolds/Virotypes)	Simpson (Scaffolds/Virotypes)	Reference ^1^
Sv2475.1d	4,348,746	637,148 (14.7%)	404	318 (78.7%)	168	408/173	407/173	4.5/3.2	0.97/0.91	This study
Sv2478.2h	3,574,388	681,061 (19.1%)	417	325 (77.9%)	171	417/172	418/173	4.8/3.2	0.98/0.87	This study
LBw.4g	9,477,618	805,244 (8.5%)	428	338 (79.0%)	183	429/186	433/187	5.1/4.1	0.99/0.97	This study
I.basta.d	15,774,944	788,317 (5.0%)	208	161 (77.4%)	98	211/99	213/99	4.3/3.2	0.98/0.93	-
I.basta.h	17,123,842	1,307,528 (7.6%)	225	175 (77.8%)	109	228/110	231/111	4.1/3.1	0.97/0.92	[22] ^2^
I.basta.md	15,050,992	711,400 (4.7%)	190	147 (77.4%)	89	190/89	192/91	4.3/3.2	0.98/0.93	-
I.basta.nd	15,377,078	770,232 (5.0%)	197	156 (79.2%)	99	197/100	198/101	4.3/3.3	0.98/0.94	[22]
GBR.sw	9,359,144	1,107,475 (11.8%)	384	297 (77.3%)	178	405/182	407/185	4.7/3.9	0.98/0.96	-

^1^ Unpublished datasets are marked with “-”. ^2^ Partially published dataset (see Table 1 for details).

**Table 3 microorganisms-10-00480-t003:** The main differences between diseased and healthy specimens of Baikal sponges.

Over-Represented Viral Taxa or Functions	Baikal Water vs. Sponges		Healthy vs. Diseased	
	LBw.4 g	*B. bacillifera*	2478.2 h	2475.1 d
Families	*Siphoviridae*	*Podoviridae*, *Lavidaviridae*, *Mimiviridae*, *Baculoviridae*, *Bicaudaviridae*, *Herelleviridae*	*Siphoviridae*, *Phycodnaviridae*, *Mimiviridae*, *Baculoviridae*, *Bicaudaviridae*, *Herelleviridae*	*Myoviridae*
Virotypes	*Arthrobacter phage Decurro*, *Synechococcus* phages *S-SKS1* and *ACG-2014h*, *Prochlorococcus phage P-GSP1*, *Xylella* phages *Sano* and *Xfas53*	*Cellulophaga* phages, *(phi38:1*, *phi10:1,* and *phi19:3)*, *Yellowstone lake virophage* 6 and others	*Nonlabens phage P12024S*, *Gordonia* phages (*GMA1*, *Wizard*, etc.), *Synechococcus phage S-CBS2*	*Enterobacteria phage Sf101*, *Bdellovibrio phage phi1422*, *Croceibacter phage P2559Y*
Putative hosts	Cyanobacteria, Actinobacteria, Firmicutes, Euryarchaeota	Bacteroidetes, Crenarchaeota, Amoebozoa, Arthropoda, Mimiviridae	Chlorophyta, Amoebozoa,	Proteobacteria
Functional categories (except “Metabolism”)	‘Replication and repair’, ‘Environmental adaptation’, ‘Digestive system’	‘Folding, sorting and degradation’, ‘Translation’, ‘Cell growth and death’, ‘Aging’	‘Signal transduction’	‘Cell growth and death’
Metabolic functions	‘Metabolism of terpenoids and polyketides’, ‘Glycan biosynthesis and metabolism’, ‘Carbohydrate metabolism’	‘Nucleotide metabolism’, ‘Xenobiotics biodegradation and metabolism’	Almost all (except ‘Amino acid metabolism’), ‘Folate biosynthesis’	‘Amino acid metabolism’, ‘Riboflavin metabolism’

## Data Availability

Unprocessed virome reads for the samples of *B. bacillifera* Sv2475.1d, Sv2478.2h, and the Baikal water Lbw.4g were submitted to the National Center for Biotechnology Information (NCBI), Sequence Read Archive (SRA) database (BioProject PRJNA577390, BioSamples SAMN13025046, SAMN13025227, and SAMN16330433) [25,43]. The direct URL to the data is as follows: https://www.ncbi.nlm.nih.gov/sra/PRJNA577390 (accessed on 20 December 2021).

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
