# Peer review of "Metagenomic Assessment of DNA Viral Diversity in Freshwater Sponges, Baikalospongia bacillifera"

_microorganisms, 2022, doi:10.3390/microorganisms10020480_

Round 1

Reviewer 1 Report

Metagenomic approaches produce huge information that sometimes is difficult to be managed. In this sense, the authors of the current study provided a good effort to explain the presence of different viral taxa in the analysed sponges. However, there are some findings that should be more interesting to be presented in another way and discussed better. Particularly, since the authors compare the DNA viral genome of a healthy with a non healthy individual, they should prepare a Table with the most important differences in quantities and determination of viruses hosting the healthy with the non healthy individual. Alternatively, this could be presented in a new heat map. This way, the info would be summarised and more easily found by the reader.

Also, a microscope (or macroscopic) figure showing the necrosis lesions in the non healthy sponge would be appreciated and valuable.

Regarding the non target bacteria could a blast search identify some of the produced sequences at genus level?

In the discussion, an interesting comparison would additionally be to compare the diversity of DNA viruses obtained from this study with RNA viruses hosting sponges from other studies.

Some minor comments:

Line 30: The term “amazing” is a little deviating. I suggest to justify or delete it.

Line 32: The term “highly endemic” is not totally correct. Maybe replace with “endemic in several locations”.

Line 32: Again “combine” does not fit here very well. Maybe choose a better term such as “accommodate”

I suggest to transfer the paragraph in lines 67-73 after line 42, before the parts for viruses. It refers to the freshwater sponges and therefore fits better there, where also the biology of sponges is referred.

The section 3.3 should probably be renamed, since the largest part refers to bacterial and other non virus groups

Reviewer 2 Report

The study investigated the viral diversity of freshwater samples and healthy and diseased sponges from Lake Baikal. Considering the lack of virome information from sponges, especially in fresh water environments, this study provides valuable information and dataset for future research. I have some concerns about the writing and methods.

Major concerns:

  1. The MS is too long. Although the journal may not limit the length of MS, the unnecessary and too detailed data description leads to unfriendly reading. A lot of results, tables, figures and discussion may be moved to supporting materials. Please just present the most important findings. Usually, such a story needs only 10 printing pages.
  2. The authors compared the viromes from the lake environment to the viromes from seawater and marine sponges. I am not sure whether this comparison is meaningful. Symbiotic bacteria and viruses in sponges and other animals are kind of species specific. The difference observed in this study is due to different species or different environment?
  3. In addition to general diversity analysis, the authors may consider to analyze the host-phage interaction (see Zhou K, Qian PY, Zhang T, Xu Y & Zhang R (2021) Unique phage–bacterium interplay in sponge holobionts from the southern Okinawa Trough hydrothermal vent. Environmental Microbiology Reports 13: 675-683).

Other Comments:

Line 108: Would the centrifugation at 16000 g for 30 min precipitate (large) viral particles?

Line 287: DNA was extracted from viral particles. However, only a small number of scaffolds (673 out of 2916) have been identified as viral sequences? How about the majority of assembled sequences?

Line 589-606: This paragraph is more likely to explain why apply cross-assembly approach. Recommend moving it to the M&M section. In addition, please clearly how to "correctly use the scaffolds that". And, chimerization can be problematic between closely related species, leading to more chimerical contigs. For this, cross-assembly may assemble longer but incorrect virus scaffolds.

Line 676: What are the patterns? Please specify.

Line 719: What are the roles of viruses in impacting bacterial hosts?

Tables 1 and 2: add a citation to the downloaded datasets to distinguish between your data and previously generated data.

Round 2

Reviewer 1 Report

The quality of the manuscript has been improved and therefore I suggest publication in the present form

Author Response

Dear Reviewer,

Thank you again for your careful review and valuable comments on our study.

With best regards,

Tatyana Butina and Co-authors.

Reviewer 2 Report

  • My concern about the comparison between freshwater and marine samples remains. My question is that why you compare to Ianthella basta sponges. I am not a sponge biologist. So I am wondering whether Ianthella basta and Baikalospongia bacillifera are comparable.
  • Other responses are fine.
